# New Insights into the Behavior of NHC-Gold Complexes in Cancer Cells

**DOI:** 10.3390/pharmaceutics15020466

**Published:** 2023-01-31

**Authors:** Giuseppa Augello, Antonina Azzolina, Filomena Rossi, Filippo Prencipe, Giuseppe Felice Mangiatordi, Michele Saviano, Luisa Ronga, Melchiorre Cervello, Diego Tesauro

**Affiliations:** 1Institute for Biomedical Research and Innovation (IRIB), CNR, Via Ugo LaMalfa 153, 90146 Palermo, Italy; 2Department of Pharmacy and Interuniversity Research Centre on Bioactive Peptides (CIRPeB), University of Naples “Federico II”, Via Montesano 49, 80131 Naples, Italy; 3Institute of Crystallography (IC), CNR, Via Amendola 122/O, 70126 Bari, Italy; 4Institute of Crystallography (IC), CNR, URT Caserta, Via Vivaldi 43, 81100 Caserta, Italy; 5Institut des Sciences Analytiques et de Physico-Chimie Pour l’Environnement et les Matériaux, Université de Pau et des Pays de l’Adour, E2S UPPA, CNRS, IPREM, 64053 Pau, France

**Keywords:** NHC-gold(I) complexes, TrxR1 activity, antiproliferative effect, cancer cells

## Abstract

Among the non-platinum antitumor agents, gold complexes have received increased attention owing to their strong antiproliferative effects, which generally occur through non-cisplatin-like mechanisms of action. Several studies have revealed that many cytotoxic gold compounds, such as N-heterocyclic carbene (NHC)-gold(I) complexes, are potent thioredoxin reductase (TrxR) inhibitors. Many other pathways have been supposed to be altered by gold coordination to protein targets. Within this frame, we have selected two gold(I) complexes based on aromatic ligands to be tested on cancer cells. Differently from bis [1,3-diethyl-4,5-bis(4-methoxyphenyl)imidazol-2-ylidene]gold(I) bromide (Au4BC), bis [1-methyl-3-acridineimidazolin-2-ylidene]gold(I) tetrafluoroborate (Au3BC) inhibited TrxR1 activity in vitro. Treatment of Huh7 hepatocellular carcinoma (HCC) cells, and MDA-MB-231 triple-negative breast cancer (TNBC) cells, with Au4BC inhibited cell viability, increased reactive oxygen species (ROS) levels, caused DNA damage, and induced autophagy and apoptosis. Notably, we found that, although Au3BC inhibited TrxR1 activity, no effect on the cell viabilities of HCC and BC cells was observed. At the molecular level, Au3BC induced a protective response mechanism in Huh7 and MDA-MB-231 cells, by inducing up-regulation of RAD51 and p62 protein expression, two proteins involved in DNA damage repair and autophagy, respectively. *RAD51* gene knock-down in HCC cells increased cell sensitivity to Au3BC by significant reduction of cell viability, induction of DNA damage, and induction of apoptosis and autophagy. All together, these results suggest that the tested NHC-Gold complexes, Au3BC and Au4BC, showed different mechanisms of action, either dependent or independent of TrxR1 inhibition. As a result, Au3BC and Au4BC were found to be promising candidates as anticancer drugs for the treatment of HCC and BC.

## 1. Introduction

Many weapons are used to fight diseases, among which an important role is played by metal drugs [1,2]. Platinum complexes are widely used to treat tumor pathologies, acting as chemotherapeutics, although severe side effects occur during the therapies and cancerous cells can develop resistance [3]. In order to overcome these drawbacks, researchers have designed, synthesized, and tested novel metallodrugs focusing on different transition ion complexes [4,5]. The ability of transition metals to react with biological molecules is due to their ability to act as Lewis acids, which coordinate molecules bearing lone pair electrons. Among these elements, the gold (I) ion is one of the most promising [6]. The first gold complex approved by the FDA, in 1985, was [(2,3,4,6-tetra-*O*-acetyl-1-(thio-κS)-β-D-glucopyranosato)(triethylphosphane)gold(I)] (auranofin), initially approved as an anti-arthritis agent [7]. Auranofin was repurposed in several clinical trials and completed Phase 2 trials for prolymphocytic leukaemia (PLL)/small lymphocytic lymphoma/chronic lymphocytic leukemia (CLL) treatments [8]. One of the major differences between the antitumor activity of platinum complexes and gold compounds lies in their different biological targets. Cisplatin, and its analogues, acts through a direct interaction with DNA [3], whereas the antiproliferative activity of gold complexes is believed to affect the functionality of proteins, usually residing in mitochondria [6]. According to its main mechanism of action, auranofin manipulates the redox milieu by inhibiting redox enzymes [9], and is a potent inhibitor of thioredoxin reductase (TrxR) [6].

It is well known that TrxR, a member of the selonoproteins family, is part of the thioredoxin (Trx) system, consisting of Trx, TrxR, and nicotinamide adenine dinucleotide phosphate (NADPH). This complex system plays a key role in the regulation of intracellular redox homeostasis, thus protecting cells from oxidative stress damage. Increasing evidence has shown that TrxR is involved, and overexpressed, in various pathological processes, such as tumorigenesis and tumor progression [10]. TrxR1 overexpression correlates with poor prognoses and lower survival-rate in patients with different types of cancers, including (hepatocellular carcinoma) HCC and breast cancer (BC) [11,12,13,14]. Therefore, TrxR1 is considered a promising target for anticancer therapies, and there is an increasing interest in developing effective inhibitors of this enzyme [15].

The promising chemotherapeutic effects of auranofin pushed researchers to extend research to other compounds, designing new bicoordinate Au (I) complexes. In light of this challenge, N-heterocyclic carbene (NHC) ligands were found to give rise to strong carbon-to-metal bonds, resulting in highly robust complexes in physiological conditions [16,17].

Indeed, several studies were carried out on NHC-Au complexes to find other available targets [18,19]. On the whole, in recent years a pronounced antitelomerase activity of this class of complex has been investigated [19,20]. Moreover, it was demonstrated that NHC gold complexes mediate S-phase arrest via down-regulation of cyclin A, cyclin B1, and cdk2 in U-87MG human glioblastoma cells [21]. In this context, in order to improve and investigate the cytotoxic activity of these complexes, we have chosen two bis NHC-gold complexes with different structures and tested their activity at suppressing TrxR in different cancer cell lines. 

Specifically, we selected two (NHC)_2_-gold(I) complexes, bis[1-methyl-3-acridineimidazolin-2-ylidene]gold(I) tetrafluoroborate (Au3BC) [22] and bis [1,3-diethyl-4,5-bis(4-methoxyphenyl)imidazol-2-ylidene]gold(I) bromide (Au4BC) [23], and evaluated their specific anticancer properties in HCC and BC cell lines.

## 2. Materials and Methods

Solvent and chemical reagents were purchased and used as received without further purification (Sigma-Aldrich, Steinheim, Germany); ^1^H NMR spectra were acquired with Brucker 400 MHz and with Brucker 700 MHz, ESI mass spectra were recorded in positive mode with an Applied Biosystems mass spectrometer equipped with a triple quadrupole mass analyzer. The compounds 1,3-diethyl-4,5-bis(4-methoxyphenyl)imidazole bromide and bromo[1,3-diethyl-4,5-bis(4-methoxyphenyl)imidazolin-2-ylidene]gold(I) were prepared as described in the literature [24,25]; bis[1-methyl-3-acridineimidazolin-2-ylidene]silver(I) tetrafluoroborate was prepared adapting reported methods [22].

### 2.1. Synthesis of Compound Bis[1-methyl-3-acridineimidazolin-2-ylidene]gold(I) Tetrafluoroborate (Au3BC)

Bis[1-methyl-3-acridineimidazolin-2-ylidene]silver(I)tetrafluoroborate (106 mg, 0.150 mmol) was added to a solution of [AuClS(CH_3_)_2_] (44 mg, 0.150 mmol) in dichloromethane (20.0 mL), and stirred at room temperature for 5 h. The solution was filtered through a pad of celite to remove the whitish AgCl precipitate and the filtrate was reduced to minimum volume under vacuum. The product was precipitated with Et_2_O as a yellow solid. (67 mg, yield: 55.5%). 

^1^H NMR (DMSO-d_6_, 400 MHz,): δ 8.25 (d, ^3^*J* = 8.70 Hz, 2H), 7.85 (d, *J* = 7.60 Hz, 2H), 7.81–7.79 (m, 2H), 7.48 (d, *J* = 7.52 Hz, 2H), 7.23 (d, ^3^*J* = 8.62 Hz), 3.39 (s, 3H). ESI^+^-MS, m/z: 715 [M]^+^.

### 2.2. Synthesis of Compound Bis[1,3-diethyl-4,5-bis(4-methoxyphenyl)imidazol-2-ylidene]gold(I) Bromide (Au4BC)

K_2_CO_3_ (18 mg, 0.13 mmol) was added to a solution of 1,3-diethyl-4,5-bis(4-methoxyphenyl)imidazole bromide (41 mg, 0.1 mmol) and bromo[1,3-diethyl-4,5-bis(4-methoxyphenyl)imidazolin-2-ylidene]gold(I) (61 mg, 0.1 mmol) in acetone (5 mL). The resulting mixture was stirred at room temperature for 24 h. Methanol was added, and the mixture was filtered through a pad of celite. The solvent was removed under reduced pressure. Purification of the residue on silica gel (CH_3_Cl/MeOH 9.7:0.3 *v*/*v* and then 9/1 *v*/*v*) afforded the product as a white solid (61 mg, yield: 64%).

^1^H NMR (DMSO-d_6,_ 700 MHz,) δ 7.33 (d, J = 8.7 Hz, 8H), 6.99 (d, J = 8.8 Hz, 8H), 4.16 (q, J = 7.2 Hz, 8H), 3.77 (s, 12H), 1.34 (t, J = 7.2 Hz, 12H). (ESI+ -MS, m/z:) = 869.5 [M^+^-Br].

### 2.3. Measurement of TrxR Activity

TrxR1 activity was assessed using the Thioredoxin Reductase Assay Kit according to the manufacturer’s instructions (Sigma-Aldrich, Milan, Italy). TrxR1 activity was spectrophotometrically assayed at 412 nm absorbance using a microplate reader (Tecan Trading AG, Switzerland). The assay is based on the NADPH-dependent reduction of the substrate, reacting with 5,5-dithio-bis(2-nitrobenzoic acid) (DTNB). Different doses of Au3BC, Au4BC, and auranofin (positive control inhibitor) were incubated with TrxR1 extracted from rat liver. Furthermore, 10 μg of protein extracted from Huh7 cells was used for determination of TrxR1 activity in cell lysates. 

### 2.4. Measurement of Trx1 Activity

Trx1 activity was assessed using the ProteoStat™ Thioredoxin-1 assay kit according to the manufacturer’s instructions (EnZo Life Science, Plymouth Meeting, PA, USA). In brief, various doses of Au3BC, Au4BC, auranofin, and aurothiomalate (positive control inhibitor) were incubated with human recombinant Trx1 and added to each well in 96-well microtiter plates, in the presence of insulin and DTT at 25 °C for 30 min. The fluorescence intensity of each well was determined using a fluorescence reader GloMax^®^ Discover Microplate (Promega, Madison, WI, USA), using an excitation setting of 500 nm and emission filter of 603 nm.

### 2.5. Cell Lines and Cell Culture

The human hepatocarcinoma Huh7 and human MDA-MB-231 triple-negative breast cancer (TNBC) cell lines used in this study were maintained in high-glucose DMEM medium containing 10% (*v*/*v*) Fetal Bovine Serum (FBS). MDA-MB-231 cells were purchased from the American Type Culture Collection (Rockville, MD, USA), while Huh7 cells were a gift from Professor M. Levrero (Sapienza University of Rome, Italy). All cell lines were authenticated by short tandem repeat (STR) profiling (BMR Genomics, Padua, Italy).

### 2.6. Cell Viability

Cells were seeded in 96-well plates, and after 24 h were exposed to auranofin (Sigma-Aldrich), Au3BC, Au4BC, plus the antioxidant N-acetyl-L-cysteine (NAC) (Sigma-Aldrich). At the end of the treatment, MTS assays were performed using the CellTiter Aqueous OneSolution kit (Promega) according to the manufacturer’s instructions. Cell viability was expressed as a percentage of the absorbance measured in the control cells. Values were expressed as the means ± Standard Deviation (SD) of three separate experiments, each performed in triplicate.

### 2.7. Intracellular ROS Detection

Cells were seeded in 96-well plates and after 24 h, were exposed to Au4BC or hydrogen peroxide (H_2_O_2_). The cells were washed with phosphate-buffered saline (PBS) prior to probing with 40 μM H_2_DCFDA (Invitrogen, Milan, Italy) for 1 h at 37 °C with 5% CO_2_ in the dark. Excess H_2_DCFDA was removed with PBS. The fluorescence intensity was measured using a GloMax^®^ microplate reader, with excitation and emission wavelengths of 485 nm and 530 nm, respectively. The fluorescence intensity of dichlorofluorescein (DCF) was calculated as the fluorescence intensity of treated cells relative to non-treated cells.

### 2.8. Western Blotting

RIPA buffer (Cell Signaling Technologies Inc., Beverly, MA, USA) was used to obtain cell lysates, and Western blotting was performed as previously described [26], with primary antibodies raised against PARP1, RAD51, γ-H2AX, p62, and LC3 (Cell Signaling Technologies, Beverly, MA, USA), and β-actin (Sigma-Aldrich). The relative expression levels were calculated as the ratio of drug-treated samples versus control (DMSO) and corrected using the quantified level of β-actin expression. Original immunoblotting images are shown in Appendix A.

### 2.9. Cell Transfection

For small interference (siRNA) transfection, 3.5 × 10^5^ Huh7 cells were seeded in 6-well plates in medium without antibiotics. After 24 h, 50 nmol/L of RAD51 siRNA (siRAD51) were used for gene silencing. Control cell transfection was performed with a Negative Control siRNA (siNC). siRAD51 (SI02663682) and siNC (1027281) were purchased from QIAGEN (Germantown, MD, USA). Cell transfections were carried out with Lipofectamine RNAiMax (Invitrogen), following the manufacturer’s instructions. After 24 h, the cells were detached and seeded in 96-well plates for an MTS assay, or in 6-well plates for protein extraction.

### 2.10. Statistical Analyses

Statistical analysis was performed using the Student’s *t*-test. The criterion for statistical significance was *p* < 0.05.

## 3. Results

### 3.1. Selection and Synthesis of Gold Complexes

We selected two gold complexes from the literature: bis[1-methyl-3-acridineimidazolin-2-ylidene]gold(I) tetrafluoroborate (Au3BC) and bis [1,3-diethyl-4,5-bis(4-methoxyphenyl)imidazol-2-ylidene]gold(I) bromide (Au4BC) (Figure 1 right). Au3BC contains acridine-based NHC ligands (Figure 1 left).

The acridine moiety is well known as a DNA intercalator, therefore this gold NHC complex could have an additional biological preference to interact specifically with DNA, being endowed with bifunctional functions. Previous fluorescence studies have also demonstrated that, in two highly resistant cancer cells (i.e., A549 and MiaPaca2), the complex is mainly contained in the lysosomes rather than in the nucleus [27]. Therefore, it is worth obtaining additional data about the action and efficacy of this complex on the viability of other cell lines. The second complex, Au4BC, has displayed cytotoxic activity on MCF-7 and MDA-MB-231 breast cancer, as well as on HT-29 colon cancer, cell lines [23]. However, previous literature studies reported that it was nearly inactive against TrxR (IC_50_ > 10 μM), therefore its mode of action is still unclear, and it needs further investigation [23].

The selected complexes were synthesized following and/or adapting described synthetic procedures [22,23]. Above all, Au4BC was obtained by adding bromo[1,3-diethyl-4,5-bis(4-methoxyphenyl)imidazolin-2-ylidene]gold(I) to one equivalent of the proligand in acetone solution (Figure 2). After that, the complex was purified by silica gel chromatography and later crystallization, it was identified by ^1^H NMR and ESI MS experiments.

### 3.2. Biological Assays

#### 3.2.1. Thioredoxin Reductase 1 (TrxR1) Activity

We investigated the inhibitory effectiveness of Au3BC and Au4BC on TrxR1 activity. Different concentrations of Au3BC, Au4BC, and auranofin, the latter used as a TrxR1 inhibitor control, were incubated with NADPH-reduced recombinant rat liver TrxR1 enzyme, and then TrxR1 activity was determined by a DTNB reduction assay. Auranofin and Au3BC exhibited strong inhibitory activity on TrxR1, while at the same dose Au4BC did not inhibit TrxR1 activity (Figure 3A).

In addition, we also evaluated the inhibitory activity of Au3BC and Au4BC in protein samples extracted from Huh7 cells. As shown in Figure 3B, auranofin and Au3BC, but not Au4BC, exhibited significant inhibitory activity on TrxR1 present in the cell lysate.

Furthermore, to investigate if Au3BC and Au4BC may modulate the thioredoxin1 (Trx1) enzymatic activity, they were tested in vitro using a Trx1 assay kit. The results obtained showed that neither compound modified Trx1 activity (Appendix A).

#### 3.2.2. Effect of Au3BC and Au4BC on Cell Viability of HCC and BC Cell Lines

Subsequently, we examined the effect of Au3BC and Au4BC on the cell viability of the Huh7 HCC cell line and the MDA-MB-231 triple-negative breast cancer (TNBC) cell line. As shown in Figure 4, auranofin and Au4BC significantly reduced the cell viability, in a dose-dependent manner, in both cell lines, while no effect was observed after treatment with Au3BC.

#### 3.2.3. Effect of Au3BC and Au4BC on the Activation of the Apoptosis and Autophagy in HCC and BC Cells

To assess the effect of Au3BC and Au4BC on the activation of apoptosis and autophagy, Western blot analyses were conducted, and the modulation of the proteins involved in these processes evaluated. Western blot analyses of Huh7 and MDA-MB-231 cell lysates showed cleavage of Poly(ADP-ribose)polymerase 1 (PARP1) after treatment with Au4BC, indicating the activation of apoptotic responses in both cell lines (Figure 5). Furthermore, treatment with Au4BC determined a decrease in p62 protein expression levels, and an increase in the lipidated form of light-chain protein 3 (LC3II), indicating the activation of an autophagic flux in the Huh7 and MDA-MB-231 cells (Figure 5).

In contrast, Au3BC did not induce cleavage of PARP1 in any of the cell lines, but induced autophagic flux blockage, as demonstrated by the increase of p62 expression levels in both cell lines (Figure 5). 

Furthermore, Au4BC treatment induced a strong increase in phospho-H2AX (γ-H2AX) histone expression levels, a marker of DNA damage, and a decrease of the DNA repair protein RAD51, showing that Au4BC treatment induced DNA lesions in Huh7 and MDA-MB-231 cells.

Modulation of γ-H2AX levels was not observed in cells treated with Au3BC. Instead, a strong increase in RAD51 expression levels was observed after treatment with Au3BC in both cell lines. These results suggest that cells were resistant to DNA damage induced by Au3BC treatment (Figure 5).

#### 3.2.4. Au4BC Induced Intracellular Reactive Oxygen Species (ROS) Production

Since high levels of ROS are known to produce DNA damage, we next evaluated the effect of Au4BC treatment on the production of reactive oxygen species (ROS), using the cell-permeable fluorescent probe H_2_DCFDA. As shown in Figure 6A, Au4BC treatment for 48 h significantly increased intracellular ROS levels in the Huh7 and MDA-MB-231 cells.

To determine whether Au4BC affected cell viability via ROS generation, we tested the effects of the ROS scavenger N-acetyl-L-cysteine (NAC) on cell viability. Cells were pre-treated with NAC (5 mM) for 2 h and subsequently treated with different concentrations of Au3BC for an additional 24 h in the presence of NAC. The results, shown in Figure 6B, demonstrated that NAC significantly restored cell viability in Au4BC treated cells to a level similar to that of the control (Figure 6B). These results indicate that Au4BC induced cancer cell death by regulating intracellular ROS.

#### 3.2.5. siRNA-Mediated Knockdown (KD) of RAD51 Gene Expression Potentiates the Sensitivity of Huh7 Cells to Au3BC

To evaluate the functional contribution of the RAD51 increase observed after treatment with Au3BC, *RAD51* gene expression was knocked down in Huh7 cells by small interfering RNA (siRAD51), and then the cells were treated with Au3BC for 72 h. After *RAD51* gene knockdown (KD), a decrease in the RAD51 protein expression levels were confirmed by a Western blot analysis (Figure 7A). Furthermore, MTS assays showed that *RAD51* KD sensitized cells to Au3BC treatment, compared with cells transfected with control siRNA (siNC) (Figure 7B). Cells transfected with siRAD51 showed a significant reduction of cell viability after treatment with 2.5 µM and 20 µM Au3BC (15% and 30%, respectively) compared to cells transfected with siNC. In addition, *RAD51* silencing promoted Au3BC-induced PARP1 cleavage, increased γ-H2AX expression levels, and decreased p62 protein expression, the latter suggesting the completion of autophagic flux (Figure 7C). Overall, these results suggest that upon Au3BC treatment, RAD51-induced expression protected cells from Au3BC antitumor effects.

## 4. Discussion

In recent years the incidence of some types of tumors, including HCC and BC, has considerably increased in Western countries [28,29]. HCC is the sixth cause of cancer-related death worldwide and the most common primary liver malignancy, accounting for about 90% of cases [28]. BC is the first malignancy and the second leading cause of cancer-related death worldwide in women [29,30,31]. The treatment of these neoplasia with existing medical therapies (systemic chemotherapy, immunotherapy) is at present not satisfactory, and has induced researchers to identify new and more efficient antitumor agents and to develop more appropriate therapeutic approaches.

Even though there have been recent improvements in both surgical and systemic therapies, HCC and BC remain associated with poor outcomes due to their late diagnoses, and high rates of recurrence and mortality. Particularly, compared to all different types of BC, the treatment of triple-negative breast cancer (TNBC), a subtype of BC that does not express estrogen and progesterone receptors (ER and PR) and does not overexpress HER-2, remains highly challenging due to the aggressive nature of the disease and limited responses to therapies.

Several studies have revealed that many gold compounds, such as NHC-gold(I) complexes, are endowed with cytotoxic properties against several cancer cell lines [27,32,33].

In the present study, we selected two gold(I) complexes, namely Au3BC and Au4BC, and their anticancer properties were evaluated in HCC cells, characterized by an innate pharmacological resistance, and in metastatic, highly aggressive, invasive, and poorly differentiated BC cells (TNBC subtype).

Au4BC has been reported to have cytotoxic activity against breast cancer as well as colon cancer cell lines [23], however, it was nearly inactive against TrxR1. Its mode of action is therefore, still unclear, and it needs further investigation. Here, we confirmed and extended previous data analyzing the mechanism responsible of the anticancer effects using the HCC Huh7 and TNBC MDA-MB-231 cell lines. Au4BC did not inhibit TrxR1 activity in vitro up to 1.6 µM; however, other studies have reported, in different experimental conditions, an IC_50_ at a very high dose of >10 µM [23]. 

Au4BC reduced cell viability in a dose-dependent manner in HCC and BC cell lines. Mechanistically, Au4BC treatments induced apoptotic cell death and activated autophagic flux. Moreover, Au4BC treatment induced intracellular ROS production in HCC and BC cells. It is well known that high levels of ROS cause DNA damage, and we observed a strong induction of γ-H2AX levels, a marker of DNA damage, recruited in DNA double strand break (DSBs) foci to allow the assembling of repair machinery. In addition, we demonstrated that the ROS scavenger NAC significantly reversed Au4BC’s effects on cell viability, suggesting that Au4BC induced cancer cell death by regulating intracellular ROS. All these results suggest that Au4BC has a potent antitumor activity, and that within the cell it has other different targets than TrxR1.

Previous studies have demonstrated that Au3BC accumulated mainly in the lysosomes, rather than in the nucleus, of lung and pancreatic cancer cells [34]. The functional consequence of this accumulation and the effect on cancer cell viability has not been studied in detail. Therefore, it was worth investigating its mechanism of action and the efficacy of this complex on the viability of cancer cells, and particularly in very resistant cancer models, such as HCC and BC. Here, we found that Au3BC inhibited TrxR1 activity in a dose-dependent manner, although up to 20 µM it had no effects on HCC and BC cells viability, a similar behavior has been reported against lung cancer A549 cells [34]. The mechanism of this chemoresistance has not been investigated previously. We found that Au3BC treatment induced a protective response mechanism, as it did not induce apoptotic cell death and DNA damage as compared with Au4BC. Instead, Au3BC induced a pro-survival autophagic flux blockage and up-regulation of RAD51 expression, a protein that plays a key role in the DNA damage response (DDR) pathway. We demonstrated that *RAD51* gene silencing significantly sensitized cells to Au3BC treatment, induced DNA damage, promoted the completion of the autophagic process, and activated cell apoptosis. These results suggest that, upon Au3BC treatment, RAD51-induced expression protected cells from Au3BC’s antitumor effects. 

## 5. Conclusions

In summary, our results highlight the potential of the NHC-Gold complexes Au3BC and Au4BC in the treatment of HCC and BC, and demonstrated that the two complexes showed different mechanisms of action, either dependent or independent on TrxR1 inhibition. Unexpectedly, and of great interest, it is our observation that the inhibition of TrxR1 alone by NHC-Gold complexes, such as Au3BC, is not sufficient to induce cell death, as cancer cells activated pro-survival pathways to escape the pharmacological treatment. On the contrary, drugs that do not inhibit TrxR1, as in the case of Au4BC, might still be very useful, due to their effects on other targets, whose identification will help to develop new potential anticancer therapies. These observations therefore deserve to be further investigated.

## Figures and Tables

**Figure 1 pharmaceutics-15-00466-f001:**
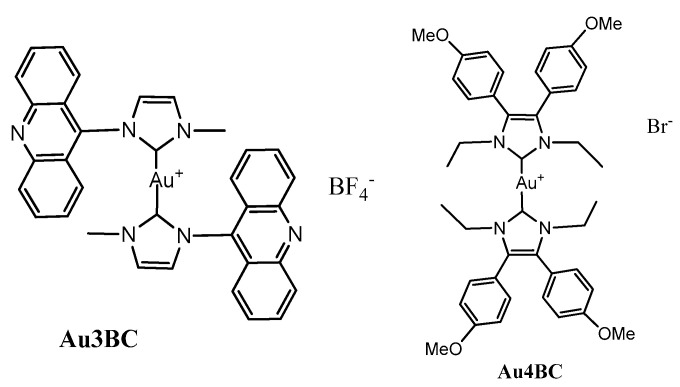
Chemical structures of the selected complexes Au3BC and Au4BC.

**Figure 2 pharmaceutics-15-00466-f002:**
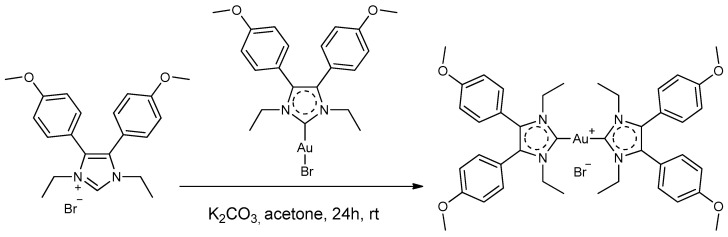
Scheme of synthetic route of Au4BC.

**Figure 3 pharmaceutics-15-00466-f003:**
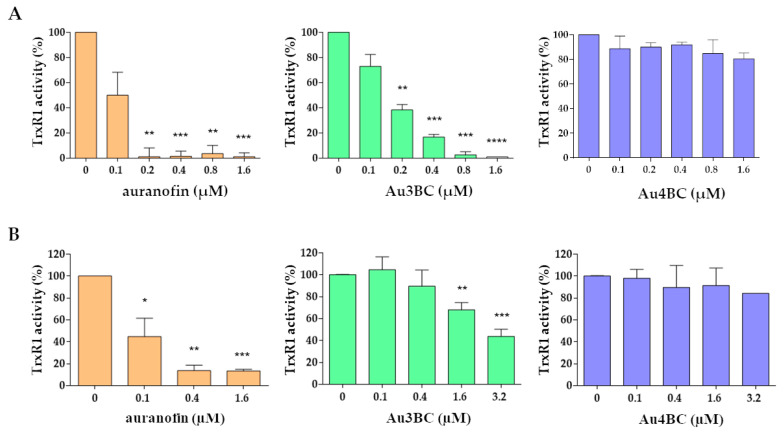
Inhibition of TrxR1 activity by Au3BC and Au4BC in vitro. (**A**) recombinant rat liver TrxR1, and (**B**) 10 µg of protein samples extracted from Huh7 cells, were incubated with different concentrations of Au3BC and Au4BC, and then TrxR1 activity was assayed by a DTNB reduction assay. Auranofin was used as the positive control inhibitor for TrxR1. * *p* < 0.05, ** *p* < 0.01, *** *p* < 0.005, **** *p* < 0.001.

**Figure 4 pharmaceutics-15-00466-f004:**
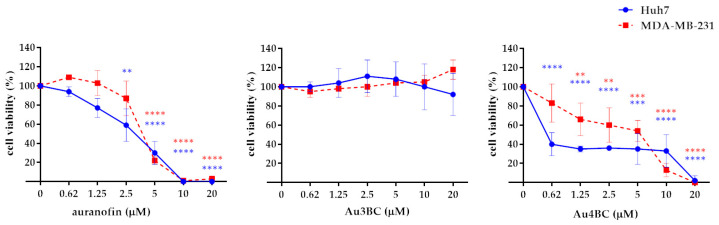
Effects of Au3BC and Au4BC treatment on cell viability in HCC and BC cell lines. Huh7 and MDA-MB-231 cells were treated with different concentrations of auranofin, Au3BC, and Au4BC for 72 h. Cell viability was determined by using an MTS assay. Auranofin was used as the positive control inhibitor for TrxR1. Data are expressed as the percentage of control cells and are the means ± SD of three separate experiments, each performed in triplicate. ** *p* < 0.01, *** *p* < 0.005, **** *p* < 0.001.

**Figure 5 pharmaceutics-15-00466-f005:**
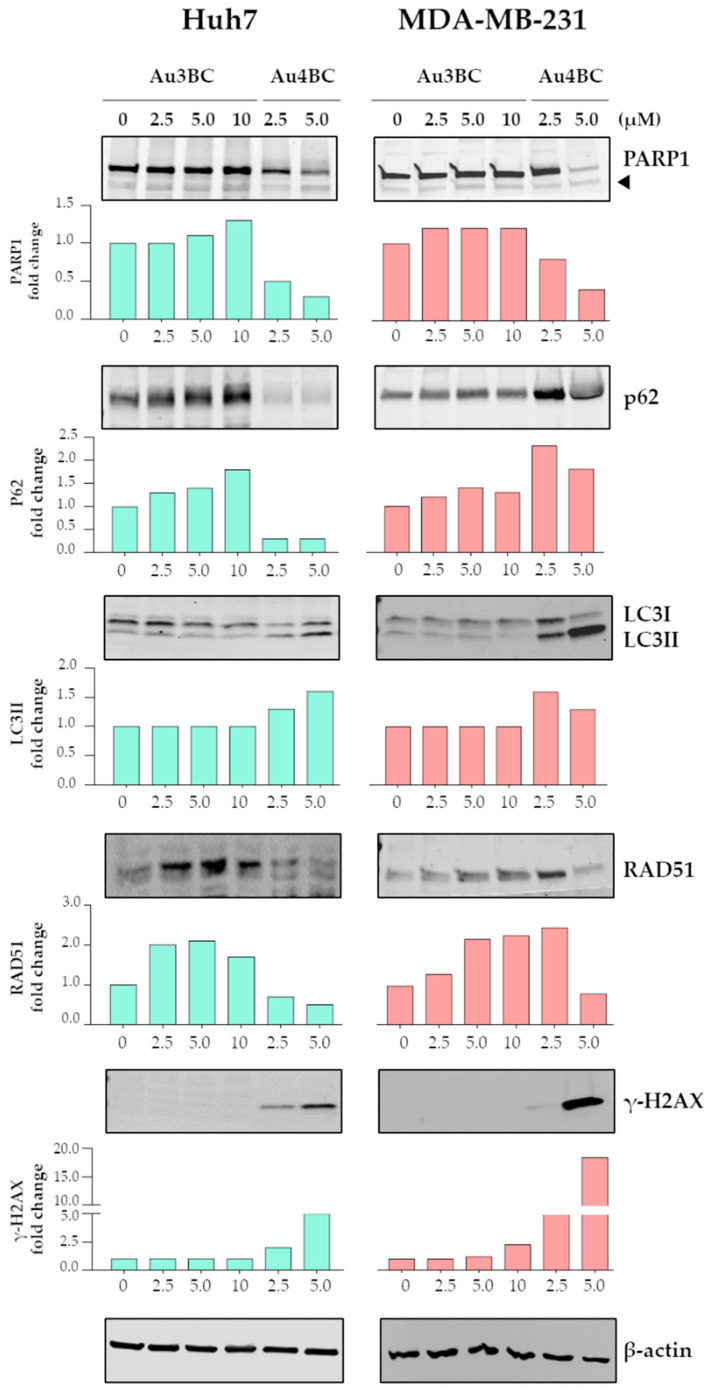
Immunoblotting evaluation of apoptosis, autophagy, and DNA damage-related protein expression in HCC and BC cells after Au3BC and Au4BC treatments. Cells were exposed to the indicated Au3BC and Au4BC concentrations for 72 h. The PARP1 fragment (85 kDa) is indicated by an arrowhead. For LC3 protein, the relative protein expression levels of LC3II were calculated. The numbers in the bar plots represent the ratio of the relevant protein normalized with β-actin, with vehicle-treated control samples arbitrarily set at 1.0.

**Figure 6 pharmaceutics-15-00466-f006:**
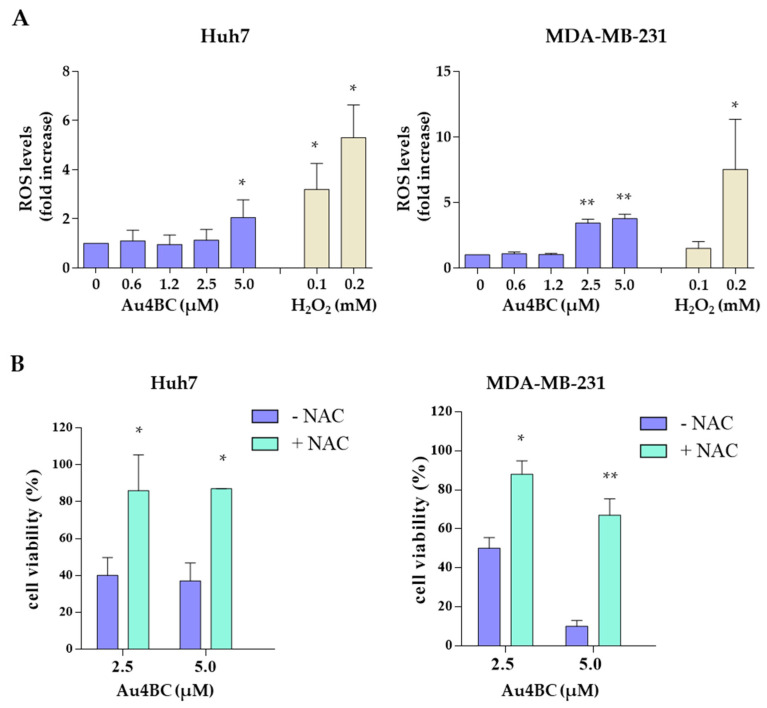
ROS generation in Huh7 and MDA-MB-231 cells treated with Au4BC. (**A**) Cells were untreated or treated with Au4BC at the indicated concentrations for 48 h, and intracellular ROS levels were evaluated using H_2_DCFDA as a probe. Hydrogen peroxide (H_2_O_2_) was used as a positive control. Data are expressed as the percentage of control cells and are the means ± SD of two separate experiments, each performed in triplicate. * *p* < 0.05, ** *p* < 0.01. (**B**) Huh7 and MDA-MB-231 cells were pre-treated for 2 h with 5 mM NAC and then treated with Au4BC for 24 h in the presence or absence of NAC. Cell viability was assessed by an MTS assay. Data are expressed as in Figure 4. * *p* < 0.05; ** *p* < 0.01.

**Figure 7 pharmaceutics-15-00466-f007:**
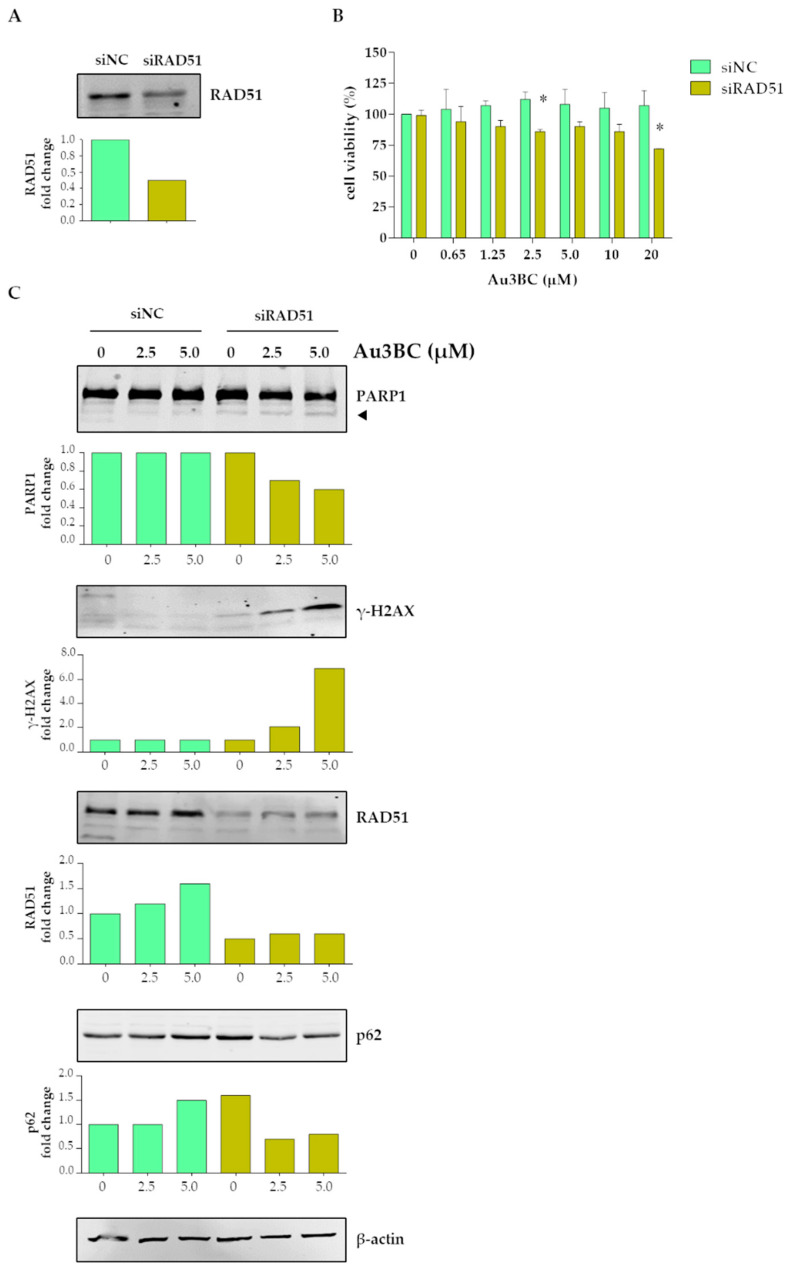
*RAD51* knockdown sensitizes Huh7 cells to Au3BC-mediated cell death. (**A**) RAD51 expression levels in cells transfected for 72 h with RAD51 siRNA (siRAD51) or with control siRNA (siNC). (**B**) Cell viability after treatment with the indicated Au3BC concentrations for 72 h of cells transfected with siRAD51 or siNC. * *p* < 0.05. (**C**) Representative Western blotting of PARP1, γ-H2AX, RAD51, and p62 levels expressed in Huh7 cells transfected with siRAD51 or siNC. The PARP1 fragment (85 kDa) is indicated by an arrowhead. The numbers in the bar plots represent the ratio of the relevant protein normalized with β-actin, with vehicle-treated control samples arbitrarily set at 1.0.

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
