# Peer review of "New Insights into the Behavior of NHC-Gold Complexes in Cancer Cells"

_pharmaceutics, 2023, doi:10.3390/pharmaceutics15020466_

Round 1

Reviewer 1 Report

 Augello et al have done very good piece of work.  They have synthesized NHC-Gold complexes, Au3BC and 34 Au4BC, and shown different mechanisms of action for these complexes.

I found this an good work and can be published in Pharmaceutics.

Author Response

Augello et al have done very good piece of work.  They have synthesized NHC-Gold complexes, Au3BC and 34 Au4BC, and shown different mechanisms of action for these complexes. I found this an good work and can be published in Pharmaceutics.

We thank the first reviewer for appreciating the manuscript.

Reviewer 2 Report

The manuscript regards the mechanism of action of two gold(I) complexes with NHC ligands. The two complexes are already known in the literature for their antiproliferative activities. The topic of the manuscript is suitable for "pharmaceutics" and interesting to be investigated, since it is commonly believed that the mechanism of action of gold(I) complexes is the inhibition of Trx. This study underlines that gold complexes can exert their anticancer activity also via other routes.

MInor points:

-The synthesis of complex Au4BC reported in Figure 2 is not clear and the conditions must be specified, in particular by adding the presence of potassium carbonate as base and the solvent. The reaction is in fact between the AuX(NHC) complex and the NHC proligand (not ligand as reported in line 195).

-I suggest the authors to check the references, since in Page 4, lines 189 and 192 ref 22 does not regard complex Au4BC (the correct reference is 23).

Author Response

The manuscript regards the mechanism of action of two gold(I) complexes with NHC ligands. The two complexes are already known in the literature for their antiproliferative activities. The topic of the manuscript is suitable for "pharmaceutics" and interesting to be investigated, since it is commonly believed that the mechanism of action of gold(I) complexes is the inhibition of Trx. This study underlines that gold complexes can exert their anticancer activity also via other routes.

Minor points:

-The synthesis of complex Au4BC reported in Figure 2 is not clear and the conditions must be specified, in particular by adding the presence of potassium carbonate as base and the solvent. The reaction is in fact between the AuX(NHC) complex and the NHC proligand (not ligand as reported in line 195).

The figure was improved adding reaction conditions

-I suggest the authors to check the references, since in Page 4, lines 189 and 192 ref 22 does not regard complex Au4BC (the correct reference is 23).

We are sorry for this mistake. As suggested we have checked and modified the reference number in the text.

Reviewer 3 Report

the manuscript is probably unbalanced between the chemical part and the biological part. The synthesis and characterization of metal complexes has already been reported in other works. The authors again described the synthesis for only one of the complexes. This part is really very confusing.

Author Response

the manuscript is probably unbalanced between the chemical part and the biological part. The synthesis and characterization of metal complexes has already been reported in other works. The authors again described the synthesis for only one of the complexes. This part is really very confusing.

We improved the chemical part. The synthesis description and the chemical characterization of Au3BC now is reported.

Round 2

Reviewer 3 Report

Line 117 correct 1H NMR – DMSO-d6

Line 118 correct ESI+-MS

In fig 2 correct Au+

Author Response

We thank the reviewer and we have corrected as suggested